# Nutritional Status of Coronary Artery Disease Patients—Preliminary Results

**DOI:** 10.3390/ijerph20043464

**Published:** 2023-02-16

**Authors:** Joanna Popiolek-Kalisz, Piotr Blaszczak

**Affiliations:** 1Clinical Dietetics Unit, Department of Bioanalytics, Medical University of Lublin, ul. Chodzki 7, 20-093 Lublin, Poland; 2Department of Cardiology, Cardinal Wyszynski Hospital in Lublin, al. Krasnicka 100, 20-718 Lublin, Poland; 3Department of Biotechnology, Microbiology and Human Nutrition, University of Life Sciences in Lublin, ul. Skromna 8, 20-704 Lublin, Poland

**Keywords:** coronary artery disease, nutritional status, nutritional risk, bioelectrical impedance analysis, cardiovascular disease, phase angle, malnutrition

## Abstract

Introduction: Malnutrition is a condition that negatively impacts the clinical outcomes of numerous diseases. The aim of this study was to assess the nutritional status of coronary artery disease (CAD) patients and to investigate its relationship with the main clinical aspects of CAD. Material and methods: 50 CAD patients undergoing coronary angiography were enrolled in this study. The nutritional status assessment was based on Nutritional Risk Score 2002 (NRS 2002), body mass index (BMI), and bioelectrical impedance analysis (BIA) measurements. Results: The analysis showed a moderate inverse correlation between NRS 2002 and BIA phase angle measured at 50 kHz (R: −0.31; *p* = 0.03) and Z_200/5_ parameter (R: 0.34; *p* = 0.02). The analysis of CAD clinical parameters showed a significant correlation between NRS 2002 and Canadian Cardiovascular Society (CCS) class (R: 0.37; *p* = 0.01). Left ventricle ejection fraction (LVEF) was correlated with BMI (R: 0.38; *p* = 0.02), however further BIA revealed hydration changes as LVEF was correlated to intracellular (ICF) and extracellular fluid (ECF) proportion: positively with ICF (R: 0.38; *p* = 0.02) and negatively with ECF (R: −0.39; *p* = 0.02). Conclusions: NRS 2002 and BIA are valuable tools for nutritional status assessment in CAD patients. Malnutrition is related to the severity of CAD symptoms, particularly in women. Maintaining proper nutritional status can have a potentially important role in this group of patients.

## 1. Introduction

Nutritional status assessment is often a neglected aspect of everyday clinical practice. According to European Society for Clinical Nutrition and Metabolism (ESPEN) criteria, malnutrition is a condition resulting from a lack of intake or uptake of nutrition which leads to altered body composition, diminished physical and mental function and impaired clinical outcome of the disease [1]. Malnutrition can accompany chronic diseases such as coronary artery disease (CAD). It is one of the factors that can potentially impact their development and treatment efficacy [2,3]. Nutritional status assessment can be based on multiple methods. They include descriptive risk scales application, body mass index (BMI) value assessment, or bioelectrical impedance analysis (BIA) which provides information regarding body composition, and direct bioelectrical parameters which are recognized nutritional status markers [4,5].

BMI is a parameter calculated by the ratio of body mass to the square of height. It is widely available and non-invasive, as it requires only a scale and stadiometer. According to the World Health Organization, BMI value can be interpreted as underweight (<18.5 kg/m^2^), normal (18.5–24.99 kg/m^2^), overweight (25.0–29.99 kg/m^2^) or obesity (≥30 kg/m^2^) [6]. According to ESPEN guidelines, BMI is a parameter used for malnutrition diagnosis (when BMI < 18.5 kg/m^2^) [1]. Despite its wide application, BMI is discussed as an insufficient parameter for nutritional status assessment, as it does not provide information about muscle or fat tissue content. Their proportion might be crucial for proper patient’s nutritional status assessment.

BIA is a non-invasive method of body composition assessment based on different electrical properties of tissues [7]. The measurement requires passing low-intensity alternating electrical current through the body and then evaluating the properties of the current coming back to the analyzer. BIA provides information about body impedance which is the combination of resistance and reactance. Resistance is the result of electrical resistivity of the low-conductive tissues, mainly fat tissue. It causes a voltage drop in the applied current. Reactance is the phenomenon caused by the electrical capacitance of cell membranes. Due to bilayer lipid structure, they are poor conductors, but on the other hand, they can accumulate electrical potential on both sides which makes them bioelectrical capacitators. Reactance causes a phase shift of the applied current, which is called the phase angle (PA). Well-nourished cells have enough resources and energy to build and maintain the proper structure of their membranes. Thus, it results in high electrical capacitance and high PA. Relatedly, malnutrition can be characterized by low capacitance and low PA [8,9,10]. PA is a well-recognized parameter of nutritional status [8]. Moreover, after applying relevant formulas, BIA provides information about the main compartments of the body: total body water (TBW), which consists of extracellular fluid (ECF) and intracellular fluid (ICF), body cell mass (BCM), fat mass (FM) and fat-free mass (FFM), which is the sum of BCM, ECF and bone mineralization [11]. FM can be used, e.g., for overweight and obesity assessment, which are the CAD risk factors [12,13,14,15,16]. Normal body fat range is based on sex and age (21–32% for women and 8–19% for men aged 20 to 39 years; 23–33% for women and 11–21% for men aged 40 to 59 years; and 24–35% for women and 13–24% for men aged over 60 years). On the other hand, FFM which consists of skeletal muscle mass is important in the context of malnutrition detection. TBW can serve as a hydration status assessment parameter. The normal range varies between 45% and 60% for adult women and 50% and 65% for adult men. As mentioned above, malnutrition could interfere with the course of the underlying disease and treatment outcomes.

Nutritional risk can be also assessed with descriptive scales, such as the Nutritional Risk Score 2002 (NRS 2002) or Subjective Global Assessment (SGA) [17,18]. NRS 2002 is based on simple questions referring to nutritional status decline, in which a patient is asked about the proceeding changes of BMI or food intake (0–3 points), and to disease exacerbation which leads to increased energy requirements (0–3 points). If the patient is older than 70 years old, they receive an additional point. A score of 3 points or more indicates the need for nutritional treatment [17]. In Polish conditions, nutritional risk assessment is compulsory during admission to the hospital.

CAD is a complex disease that is the result of atherosclerosis present in coronary arteries which transport oxygen and nutrients to the myocardium. The assessment of CAD structural advancement can be performed by imaging techniques. Coronary angiography is the most popular one [19]. It enables visualization of the stenosis in the coronary vessels and assessment of their significance and localization [20]. In the course of CAD, blood flow along the lesion in a coronary artery is reduced which results in hypoxia of the myocardium region supplied by this vessel. It leads to physical performance decline due to exertional chest pain. Clinical symptoms can be described with Canadian Cardiovascular Society grading (CCS). It is based on the threshold of the effort causing chest pain: from asymptomatic, through strenuous, moderate, or mild exertion, to angina at rest. Coronary blood flow decline results in myocardium necrosis which then can affect left ventricle ejection fraction (LVEF). LVEF reduction is part of heart failure development.

The relationship between nutritional status and CAD is complex. On one hand, obesity and fat tissue accumulation are risk factors for CAD development, while on the other hand, malnutrition can possibly aggravate the course of the disease [21,22,23,24].

The aim of this study was to assess nutritional status in CAD patients with the multiple available methods, and then to investigate if nutritional status and body composition parameters are correlated to main CAD aspects, such as the CCS class for the clinical performance assessment, a number of significantly lesioned coronary arteries as the structural CAD advancement, and LVEF as an expression of potential CAD complications.

## 2. Materials and Methods

Fifty (50) CAD patients (25 men and 25 women) undergoing routine coronary angiography were enrolled in the study between March and December 2022. The inclusion criteria were: (1) age 20–80 years; (2) written consent; (3) CAD diagnosis; (4) coronary angiography result. The exclusion criteria were: (1) age < 20 or >80 years; (2) lack of written consent; (3) abnormal body temperature or infection; (4) pregnancy; (5) amputations; (6) metal implants.

The mean age was 67.54 ± 9.04 years. The mean BMI was 29.11 ± 3.65 kg/m^2^, PA at 5 kHz was 2.70 ± 0.82°, PA at 50 kHz was 5.35 ± 1.05°, PA at 100 kHz was 5.41 ± 0.95°, PA at 200 kHz was 4.38 ± 0.86°, Z_200/5_ was 0.80 ± 0.04, Cm was 1.77 ± 0.78 nF, FFM was 71.29 ± 7.76%, FM was 28.69 ± 7.75%, TBW 52.20 ± 5.67%, ECF 44.35 ± 3.34%, ICF 55.67 ± 3.32%. The mean LVEF was 52.41 ± 16.08%. Thirty one (31) patients received 0 points in NRS 2002 score, 17 received 1 point, and two received 2 points. There were no patients who received 3 points or more. Among patients classified with 1 point, it was added due to age ≥ 70 years in all cases. In the coronary angiography, 19 patients did not have significant lesions in coronary arteries, 12 had one-vessel CAD, 8 had two-vessel CAD and 11 had three-vessel CAD.

The nutritional risk was assessed with NRS 2002. As described above, NRS 2002 is a score based on information about recent BMI changes or intake reduction, and information about concomitant diseases which could possibly lead to increased energy requirements. One additional point is given due to age ≥ 70 years. Interpretation of the NRS 2002 result is based on the total number of points. The result ≥ 3 points is interpreted as high nutritional risk and it is an indication of nutritional treatment.

Body mass was measured with WTL-150A scale (Lubelskie Fabryki Wag, Lublin, Poland) with 0.05 kg accuracy by the trained professional. The participants were permitted to wear only underwear for this measurement. The patient’s height was measured with a stadiometer with 1.0 cm accuracy by the trained professional.

The patients underwent BIA measurement with ImpediMed SFB7 (Impedimed Ltd., Brisbane, Australia) in accordance with the producer’s instructions. Before the examination, participants lied in a horizontal position for 5 min and were not allowed to drink, eat, or exert any physical effort in the preceding three hours. The used system was based on four self-adhesive surface electrodes connected to the analyzer. Electrodes were placed on the washed and dried skin of the left wrist and left ankle. The measurements were performed in the horizontal position with limbs resting loosely at 30–45 degrees to the body. The direct bioelectrical parameters were measured at 4 main frequencies: 5 kHz, 50 kHz, 100 kHz, and 200 kHz, while body composition information was derived from bioelectrical impedance spectroscopy mode. The measurements were taken in triplicates, and then the mean values were used for statistical analysis.

Clinical performance was assessed with the CCS scoring, which is based on the information about the threshold of the effort causing angina symptoms: from asymptomatic (CCS 0), through strenuous (CCS 1), moderate (CCS 2), or mild exertion (CCS 3), to angina at rest (CCS 4). The number of significantly lesioned coronary arteries was derived from coronary angiography. The results were defined as one-, two- or three-vessel CAD or lack of significant lesions. LVEF was assessed by a qualified medical doctor with GE Vivid 7 ultrasound (General Electronics, Boston, MA, USA) with Simpson’s biplane method.

### 2.1. Statistical Analysis

Statistical analyses were performed with the RStudio software v. 4.2.0. The normality of the distribution of each parameter was checked by the Shapiro–Wilk test. The variables were presented as means (±SD).

Pearson correlation was used to analyze the linear association between selected nutritional status parameters to investigate if patients are properly assessed at the time of hospital admission. The cut-off points used for the correlation coefficient were as follows: <0.30 as low, 0.30–0.49 as moderate, and ≥0.50 as high correlation. A *p*-value below 0.05 was considered significant.

Then the patients were divided into two subgroups by clinical performance according to CCS class, i.e., CCS ≤ 2 or CCS ≥ 3. The comparison between the subgroup was performed with the Mann-Whitney-U test. A *p*-value below 0.05 was considered significant.

The patients were also divided into 5 subgroups by CCS class. The one-way ANOVA test was used to compare the BIA and anthropometrical parameters between these multiple groups. Then the linear correlation between nutritional status parameters and CCS class was investigated with the Pearson correlation test. The cut-off points used for the correlation coefficient were the same as above: <0.30 as low, 0.30–0.49 as moderate, and ≥0.50 as high correlation. A *p*-value below 0.05 was considered significant.

The correlation was also investigated between nutritional status parameters and the number of lesioned coronary vessels or LVEF. The cut-off points used for the correlation coefficient were the same as above: <0.30 as low, 0.30–0.49 as moderate and ≥0.50 as high correlation. A *p*-value below 0.05 was considered significant.

Finally, the patients were divided into subgroups by sex (men and women) in which the mentioned analyses were also performed. The subgroup included equal numbers of women (n = 25) and men (n = 25). Women were older (mean age: 69.3 ± 9.6 years) than men (mean age: 65.8 ± 8.3 years). Women had less advanced CAD (mean of 0.96 ± 1.06 significantly lesioned coronary arteries in women vs. 1.48 ± 1.26 in men), but they had more severe symptoms (mean CCS score 2.00 ± 1.15 in women vs. 1.92 ± 0.95 in men). In terms of nutritional status women were characterized by lower nutritional risk (mean NRS 2002 0.40 ± 0.50 points in women vs. 0.44 ± 0.65 points in men), higher BMI (29.72 ± 4.17 kg/m^2^ in women vs. 28.55 ± 3.08 kg/m^2^ in men) and lower PA measured at 50 kHz (5.12 ± 1.12° in women vs. 5.58 ± 0.95° in men).

### 2.2. Ethical Concerns

The study was approved by the local Bioethics Committee of the Medical University of Lublin (consent no. KE-0254/9/01/2022). The study was conducted in line with the directives of the Declaration of Helsinki on Ethical Principles for Medical Research. All participants signed a written consent agreement.

## 3. Results

### 3.1. Nutritional Status of CAD Patients

Nutritional status assessment with different approaches revealed a moderate inverse correlation between NRS 2002 scoring and PA measured at 50 kHz values (R: −0.31; 95% CI: −0.544 to −0.038; *p* = 0.03), and a positive correlation between NRS 2002 and Z_200/5_ parameter (R: 0.34; 95% CI: 0.065 to 0.563; *p* = 0.02). BMI value correlated moderately with PA measured at 5 kHz (R: 0.31; 95% CI: 0.018 to 0.548; *p* = 0.04). The detailed results are presented in Table 1.

### 3.2. Nutritional Status and Clinical Performance

Initially, CAD patients were divided into two subgroups depending on their clinical performance, as CCS ≤ 2 or CCS ≥ 3. The comparison between these subgroups revealed significant differences in NRS 2002. The patients with more symptomatic CAD were also characterized as at higher nutritional risk compared to the less symptomatic patients (0.67 ± 0.59 points vs. 0.28 ± 0.52 points; *p* = 0.02). More symptomatic patients had also lower values of PA at 50 kHz and BMI, but these differences were not significant. It is worth noting, that the number of significantly lesioned coronary arteries and LVEF also did not significantly differ regarding symptoms level. The detailed results are presented in Table 2.

Secondly, the analysis of the nutritional status and basic imaging parameters (coronary angiography result, LVEF) was also performed among the subgroups regarding the exact CCS score. These results also confirmed significant differences in NRS 2002 among the CCS subgroups (*p* = 0.01). The relationships for other investigated parameters (BMI, PA measured at 5 kHz, 50 kHz, 100 kHz, and 200 kHz, Z_200/5_ parameter, Cm, FFM, FM, TBW, ECF, ICF, LVEF, and a number of significantly lesioned coronary arteries) were not significant. The detailed results for selected nutritional status parameters are presented in Figure 1.

### 3.3. Nutritional Status and CAD Parameters

The next step was the analysis of the linear relationship between nutritional status and main CAD parameters, i.e., clinical performance in CCS class, structural CAD advancement as a number of significantly lesioned coronary arteries, and potential CAD complications as LVEF. The results showed a significant moderate correlation between NRS 2002 and CCS class (R: 0.37; 95% CI: 0.099 to 0.586; *p* = 0.01). There were correlations between the number of significantly lesioned coronary arteries and nutritional status. LVEF was significantly moderately correlated with BMI value (R: 0.38; 95% CI: 0.057 to 0.635; *p* = 0.02). LVEF was also related to hydration status, i.e., positively to ICF (R: 0.38; 95% CI: 0.073 to 0.621; *p* = 0.02) and negatively to ECF (R: −0.39; 95% CI: −0.625 to −0.080; *p* = 0.02). The detailed results are presented in Table 3.

### 3.4. Women and Men Subgroup Analysis

The analysis within the subgroups of women and men revealed a moderate inverse correlation between NRS 2002 and PA measured at 5 kHz (R: −0.42; 95% CI: −0.698 to −0.028; *p* = 0.04) and PA measured at 50 kHz (R: −0.44; 95% CI: −0.712 to −0.056; *p* = 0.03) in men, without such relationship in women. Similarly, NRS 2002 was highly correlated with the Z_200/5_ parameter only in men (R: 0.53; 95% CI: 0.169 to 0.764; *p* = 0.01). For BMI, a moderate positive correlation was present for PA measured at 5 kHz (R: 0.43; 95% CI: 0.027 to 0.708; *p* = 0.04) and PA measured at 200 kHz (R: 0.47; 95% CI: 0.081 to 0.734; *p* = 0.02) in men. Detailed results are presented in Table 4.

The subgroup analysis of the relationship between nutritional status and main CAD parameters revealed a significant high correlation between CCS class and NRS 2002 score in women (R: 0.58; *p* = 0.003). Among women, also LVEF was highly correlated with BMI (R: 0.51; *p* = 0.04). The detailed results are presented in Table 5.

## 4. Discussion

Nutritional status is an important factor impacting disease prognosis and treatment outcome, also among cardiovascular patients [1,25]. It was shown that nutritional status is correlated with the length of hospitalization in patients with hypertension and atrial fibrillation [26,27]. It is also associated with mortality in heart failure and coronary artery disease [28,29,30,31,32]. The available studies are mostly retrospective, so they are based only on the parameters available in everyday clinical practice. This prospective study was intended to implement also other nutritional status assessment methods such as BIA. The presented study confirmed prior suggestions that BMI is not the best parameter for nutritional status assessment, as among three independent methods, the crucial results were correlated only for NRS 2002 and PA measured at 50 kHz and for NRS 2002 and the Z_200/5_ parameter. BMI was correlated only with PA measure at 5 kHz, however, this frequency is not recognized as a golden standard for nutritional status assessment. On the basis of this observation, we can suggest that NRS 2002, PA measured at 50 kHz with BIA, and the Z_200/5_ parameter are good parameters for nutritional status assessment in CAD patients. In the course of the sex subgroup analysis, it turned out that the correlation between NRS 2002 scoring and PA measured at 50 kHz was present in men, but not in women.

The next step of this study was focused on the potential relationship between nutritional status or body composition and main CAD aspects: symptoms, structural advancement, and complications. The analysis showed that nutritional risk assessed with NRS 2002 was higher in patients with more severe symptoms (higher CCS class). This relationship was present in both situations: when patients were divided into two subgroups as CCS class ≤ 2 or ≥3 and between separate CCS classes. The character of this relationship turned out to be linear. It is worth noting, that this correlation was high and present mainly in women. This results in the conclusion that nutritional status is associated with the clinical presentation of female patients. From a patient’s perspective, symptoms severity is probably the most important aspect of the course of their disease. This suggests the crucial role of maintaining proper nutritional status, also in CAD patients.

On the other hand, a meta-analysis by de Borba et al. showed that cardiovascular disease patients have lower PA measured at 50 kHz than healthy ones [33]. In the presented study PA at 50 kHz was negatively correlated with CAD symptoms severity, however, this relationship was not significant. There are not available other studies analyzing the relationship between nutritional status and symptoms severity in CAD patients. This study is the first one which analyzed this aspect of malnutrition in CAD.

Nutritional status and body composition were not related to the number of significantly lesioned coronary arteries in the presented study, this observation is in line with the fact that the symptoms of CAD do not correlate adequately with the extent of coronary lesions. The observations from other studies suggest the role of visceral fat accumulation as a good predictor for coronary calcification [34]. These differences might be caused by different methodologies, as in the present study only the number of significantly lesioned arteries was taken into count and total FM was assessed, while in the study by Xiong et al. the structural advancement of CAD was assessed with tomography calcium score and segmental BIA was performed [34]. However, the mentioned study group was also hemodialysis patients which could interfere BIA results [34]. Similarly, a higher calcification score was also observed in patients with higher ECF in patients with chronic kidney disease, while in the presented study such a relationship was not present [35]. Nutritional status assessed with the nutritional risk index, which is based on laboratory parameters was also correlated to CAD advancement assessed with SYNTAX scale [36].

Heart failure is a potential complication of myocardium ischemia due to CAD. LVEF is one of the main criteria used for heart failure detection and classification. It can be assessed with different imaging techniques and echocardiography is the most available one [37]. Overhydration can result in edema which is one of the leading symptoms of heart failure. This study showed that BMI was highly correlated with LVEF value. On the other hand, LVEF was not significantly correlated with other nutritional status parameters. As noted above, BMI does not provide information about the qualitative condition in terms of body mass, so it is not possible to exclude overhydration as a possible reason for BMI increase. This aspect can be solved with BIA and was presented in this study. The proportion between ICF and ECF turned out to be associated with LVEF: ICF positively and ECF negatively. ECF is the fluid compartment present in interstitial and intravascular space, while ICF is intracellular fluid. Fluid accumulation related to LVED decrease is mainly located in ECF, which is in line with the presented results. This study shows that BIA is a valuable tool for hydration status assessment in CAD patients. Other studies also confirm that ECW can be used as an early marker of non-clinically evident edema in pre-clinical cardiovascular disease [38].

### Limitations of the Study

This study has some limitations that should be taken into consideration. The investigated population was intended to provide preliminary findings on this topic. Thus, the provided conclusions and suggestions should be now verified in a larger group. Furthermore, in the course of this study, BIA was performed as total body measurements, because the used device did not enable an option for segmental measurements. Moreover, CAD advancement was simply assessed with the number of significantly lesioned coronary arteries in coronary angiography. The analysis including SYNTAX score could provide more detailed results. However, this is the first study that analyzed the CAD clinical severity related to nutritional status. As mentioned above, further studies are needed to support these preliminary findings.

## 5. Conclusions

This study showed that NRS 2002 and BIA are both valuable and possibly complementary tools for nutritional status assessment in CAD patients. Malnutrition was correlated with CAD symptoms severity. The subgroup analysis revealed that this relationship was present in the women subgroup. This proves the important role of maintaining proper nutritional status in CAD patients’ clinical performance, particularly in women, and the role of nutritional risk screening in everyday clinical practice. It is important as women were the ones who reported higher severity of CAD symptoms compared to men. Moreover, in the course of this study, the utility of BIA for overhydration detection in CAD patients was also confirmed.

## Figures and Tables

**Figure 1 ijerph-20-03464-f001:**
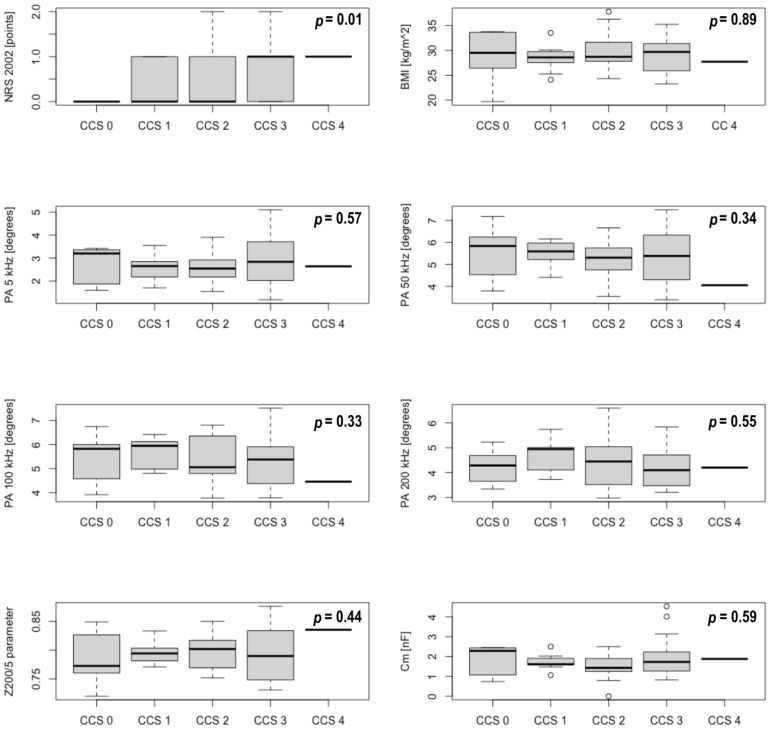
Box-plots presenting differences in nutritional parameters between consecutive CCS class subgroups.

**Table 1 ijerph-20-03464-t001:** Correlation between BMI, NRS and BIA parameters.

	R	95% CI	*p*
Parameter	NRS 2002		
BMI [kg/m^2^]	0.002	−0.288; 0.292	0.99
PA 5 kHz [°]	−0.21	−0.460; 0.074	0.15
PA 50 kHz [°]	−0.31	−0.544; −0.038	0.03 *
PA 100 kHz [°]	−0.27	−0.510; 0.009	0.06
PA 200 kHz [°]	−0.11	−0.376; 0.175	0.45
Z_200/5_ parameter	0.34	0.065; 0.563	0.02 *
Cm [nF]	−0.14	−0.407; 0.139	0.32
	BMI		
NRS 2002 [points]	0.002	−0.288; 0.292	0.99
PA 5 kHz [°]	0.31	0.018; 0.548	0.04 *
PA 50 kHz [°]	0.16	−0.139; 0.429	0.29
PA 100 kHz [°]	0.15	−0.144; 0.424	0.31
PA 200 kHz [°]	0.15	−0.151; 0.418	0.33
Z_200/5_ parameter	−0.21	−0.468; 0.090	0.17
Cm [nF]	0.19	−0.105; 0.456	0.20

PA 5 kHz—phase angle measured at 5 kHz, PA 50 kHz—phase angle measured at 50 kHz, PA 100 kHz—phase angle measured at 100 kHz, PA 200 kHz—phase angle measured at 200 kHz, Cm—capacitance of cell membranes, *—*p* < 0.05.

**Table 2 ijerph-20-03464-t002:** Comparison of nutritional status and imaging parameters between CAD with CCS class ≤ 2 or ≥3.

	CCS ≤ 2 N = 32	SD	CCS ≥ 3 N = 18	SD	*p*
NRS 2002 [points]	0.28	0.52	0.67	0.59	0.02 *
BMI [kg/m^2^]	29.28	3.86	28.79	3.33	0.76
PA 5 kHz [°]	2.61	0.63	2.87	1.08	0.61
PA 50 kHz [°]	5.40	0.90	5.26	1.31	0.54
PA 100 kHz [°]	5.49	0.83	5.27	1.16	0.27
PA 200 kHz [°]	4.46	0.88	4.24	0.84	0.30
Z_200/5_	0.79	0.03	0.80	0.05	0.85
Cm [nF]	1.65	0.59	1.98	1.02	0.52
FFM [%]	70.37	7.70	72.93	7.81	0.15
FM [%]	24.46	8.21	27.02	7.79	0.13
TBW [%]	51.51	5.63	53.42	5.70	0.13
ECF [%]	44.49	2.74	55.89	4.28	0.87
ICF [%]	55.54	2.70	44.11	4.28	0.87
LVEF [%]	51.08	16.76	54.79	15.10	0.68
Number of significantly lesioned coronary arteries	1.13	1.10	1.39	1.33	0.63

FFM—fat-free mass, FM—fat mass, TBW—total body water, ECF—extracellular fluid, ICF—intracellular fluid, LVEF—left ventricle ejection fraction, *—*p* < 0.05.

**Table 3 ijerph-20-03464-t003:** Correlation between nutritional status and main CAD parameters.

Parameter	R	95% CI	*p*
CCS class			
NRS 2002 [points]	0.37	0.099; 0.586	0.01 *
BMI [kg/m^2^]	0.02	−0.271; 0.309	0.89
PA 5 kHz [°]	0.08	−0.201; 0.352	0.57
PA 50 kHz [°]	−0.14	−0.400; 0.147	0.34
PA 100 kHz [°]	−0.14	−0.403; 0.143	0.33
PA 200 kHz [°]	−0.09	−0.357; 0.196	0.55
Z_200/5_	−0.26	−0.530; 0.063	0.11
Cm [nF]	0.08	−0.206; 0.348	0.59
FFM [%]	−0.001	−0.280; 0.277	0.99
FM [%]	−0.002	−0.279; 0.277	0.99
TBW [%]	0.002	−0.278; 0.280	0.99
ECF [%]	0.05	−0.232; 0.323	0.73
ICF [%]	−0.06	−0.328; 0.227	0.70
Significantly lesioned coronary arteries			
NRS 2002 [points]	0.01	−0.268; 0.289	0.94
BMI [kg/m^2^]	−0.13	−0.409; 0.162	0.37
PA 5 kHz [°]	0.04	−0.244; 0.312	0.80
PA 50 kHz [°]	0.002	−0.277; 0.280	0.99
PA 100 kHz [°]	0.02	−0.260; 0.297	0.90
PA 200 kHz [°]	0.09	−0.191; 0.361	0.53
Z_200/5_	0.02	−0.255; 0.301	0.87
Cm [nF]	−0.04	−0.317; 0.239	0.77
FFM [%]	0.02	−0.260; 0.297	0.89
FM [%]	−0.02	−0.294; 0.262	0.91
TBW [%]	0.02	−0.262; 0.294	0.91
ECF [%]	0.07	−0.216; 0.338	0.64
ICF [%]	−0.07	−0.340; 0.215	0.64
LVEF			
NRS [points]	0.19	−0.131; 0.479	0.24
BMI [kg/m^2^]	0.38	0.057; 0.635	0.02 *
PA 5 kHz [°]	0.12	−0.208; 0.416	0.48
PA 50 kHz [°]	0.28	−0.042; 0.545	0.09
PA 100 kHz [°]	0.28	−0.042; 0.545	0.09
PA 200 kHz [°]	0.23	−0.096; 0.506	0.17
Z_200/5_	−0.26	−0.530; 0.063	0.11
Cm [nF]	−0.01	−0.320; 0.311	0.98
FFM [%]	−0.09	−0.392; 0.235	0.60
FM [%]	0.08	−0.238; 0.389	0.61
TBW [%]	−0.08	−0.389; 0.238	0.61
ECF [%]	−0.39	−0.625; −0.080	0.02 *
ICF [%]	0.38	0.073; 0.621	0.02 *

*—*p* < 0.05.

**Table 4 ijerph-20-03464-t004:** Correlation between nutritional status parameters among women and men subgroups.

	Women			Men		
	R	95% CI	*p*	R	95% CI	*p*
NRS 2002						
BMI [kg/m^2^]	−0.05	−0.462 0.379	0.82	−0.05	−0.357; 0.448	0.80
PA 5 kHz [°]	0.05	−0.355 0.434	0.82	−0.42	−0.698; −0.028	0.04 *
PA 50 kHz [°]	−0.22	−0.566 −0.192	0.29	−0.44	−0.712; −0.056	0.03 *
PA 100 kHz [°]	−0.24	−0.580 0.171	0.25	−0.34	−0.645; 0.069	0.10
PA 200 kHz [°]	−0.19	−0.544 0.221	0.36	−0.05	−0.436; 0.352	0.81
Z_200/5_ parameter	0.18	−0.228 0.540	0.38	0.53	0.169; 0.764	0.01 *
Cm [nF]	0.09	−0.319 0.466	0.68	−0.37	−0.665; 0.034	0.07
BMI						
NRS 2002 [points]	−0.05	−0.462 0.379	0.82	0.05	−0.357; 0.448	0.80
PA 5 kHz [°]	0.32	−0.121 0.651	0.15	0.43	0.027; 0.708	0.04 *
PA 50 kHz [°]	0.22	−0.222 0.587	0.32	0.27	−0.153; 0.605	0.21
PA 100 kHz [°]	0.15	−0.292 0.536	0.51	0.34	−0.070; 0.655	0.10
PA 200 kHz [°]	−0.03	−0.448 0.394	0.88	0.47	0.081; 0.734	0.02 *
Z_200/5_	−0.30	−0.642 0.137	0.17	−0.22	−0.575; 0.198	0.29
Cm [nF]	0.34	−0.092 0.668	0.12	0.16	−0.256; 0.533	0.44

*—*p* < 0.05.

**Table 5 ijerph-20-03464-t005:** Correlation between nutritional status and main CAD parameters in women and men subgroups.

	Women		Men	
	R	*p*	R	*p*
CCS class				
NRS 2002 [points]	0.58	0.003 *	0.19	0.35
BMI [kg/m^2^]	0.11	0.63	−0.11	0.61
PA 5 kHz [°]	0.21	0.31	−0.05	0.82
PA 50 kHz [°]	−0.02	0.94	−0.30	0.15
PA 100 kHz [°]	−0.02	0.94	−0.31	0.13
PA 200 kHz [°]	−0.001	0.997	−0.20	0.35
Z_200/5_	−0.34	0.14	−0.27	0.26
Cm [nF]	0.12	0.58	0.07	0.75
FFM [%]	−0.14	0.50	0.18	0.40
FM [%]	0.13	0.52	−0.18	0.40
TBW [%]	−0.13	0.52	0.18	0.40
ECF [%]	0.05	0.81	0.10	0.64
ICF [%]	−0.05	0.81	−0.11	0.60
Significantly lesioned coronary arteries				
NRS 2002 [points]	0.27	0.20	−0.17	0.43
BMI [kg/m^2^]	−0.19	0.41	−0.03	0.88
PA 5 kHz [°]	−0.26	0.22	0.21	0.32
PA 50 kHz [°]	−0.35	0.09	0.25	0.23
PA 100	−0.28	0.18	0.23	0.28
PA 200	0.001	0.995	0.14	0.50
Z_200/5_	0.34	0.09	−0.22	0.30
Cm	−0.39	0.06	0.12	0.58
FFM	−0.31	0.14	0.06	0.77
FM	0.31	0.13	−0.06	0.77
TBW	−0.31	0.13	0.06	0.77
ECF	0.25	0.23	−0.25	0.22
ICF	−0.25	0.23	0.25	0.23
LVEF				
NRS 2002 [points]	0.15	0.52	0.23	0.34
BMI [kg/m^2^]	0.51	0.04 *	0.23	0.36
PA 5 kHz [°]	0.27	0.25	−0.01	0.97
PA 50 kHz [°]	0.35	0.13	0.35	0.14
PA 100 kHz [°]	0.33	0.15	0.37	0.12
PA 200 kHz [°]	0.22	0.36	0.31	0.20
Z_200/5_	−0.34	0.14	−0.27	0.26
Cm [nF]	0.21	0.37	−0.25	0.31
FFM [%]	−0.05	0.85	0.07	0.77
FM [%]	0.04	0.87	−0.07	0.77
TBW [%]	−0.04	0.87	0.07	0.77
ECF [%]	−0.19	0.42	−0.37	0.12
ICF [%]	0.19	0.42	0.36	0.13

*—*p* < 0.05.

## Data Availability

The data that support the findings of this study are available from the corresponding author upon reasonable request.

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
