# Peer review of "Nutritional Status of Coronary Artery Disease Patients—Preliminary Results"

_ijerph, 2023, doi:10.3390/ijerph20043464_

Round 1
Reviewer 1 Report
This study determine the nutritional status of coronary artery disease (CAD) patients and its correlation with the main clinical aspects of CAD. Generally, it is fairly written and the preliminary findings show the crucial role of nutritional status in patients' clinical performance. There are only few comments and questions for the authors, to improve the content and readability of this paper.
1) Line 30, please write the words in full first before the use of the acronym.
2) Line 101, any pre-screening test carried out before the study?
3) Line 106, "which leads could possible lead to ..." please check and correct the syntax errors throughout the manuscript.
4) Line 150-151 and line 154, normally 0.5-0.7 is moderate and beyond 0.7 only is considered as high correlation. Perhaps the authors can relook into this range of correlation?
5) Line 166, the mean age is 67.54 years, and in line 101, recruitment is open to 18-85 years. It shows that the population is more prone to be the elderly group. This may affect the findings as the health status of the elderly may not be suitable or representative enough to the 18-85 years population. The gender subgroup analysis is also carried out but basically the age is not taking into consideration.
6) Table 1,3 & 4, the values for CI is slightly confusing, the authors may put a bigger gap between or put a dash to indicate the interval.
Author Response
Dear Reviewer,
Thank you very much for your time, effort, and valuable comments on our manuscript. We corrected the manuscript according to your suggestions and we believe that it deeply improved its quality. Below you can find our responses to your comments. Thank you for this opportunity.
This study determine the nutritional status of coronary artery disease (CAD) patients and its correlation with the main clinical aspects of CAD. Generally, it is fairly written and the preliminary findings show the crucial role of nutritional status in patients' clinical performance. There are only few comments and questions for the authors, to improve the content and readability of this paper.
- Line 30, please write the words in full first before the use of the acronym.
The full name was given.
- Line 101, any pre-screening test carried out before the study?
There only pre-screening referred to inclusion and exclusion criteria identification. If a patient met inclusion and did not meet exclusion criteria, they were offered study enrollment. However, written consent was needed to perform the measurements, thus the study did not include all the patients undergoing angiography during this period, only the ones who agreed to participate.
- Line 106, "which leads could possible lead to ..." please check and correct the syntax errors throughout the manuscript.
Thank you very much for this remark. This sentence was corrected to make it clear. The syntax was also revised throughout the rest of the manuscript.
- Line 150-151 and line 154, normally 0.5-0.7 is moderate and beyond 0.7 only is considered as high correlation. Perhaps the authors can relook into this range of correlation?
The cut-off values were changed according to Reviewer's suggestion and the literature referring to medical studies.
- Line 166, the mean age is 67.54 years, and in line 101, recruitment is open to 18-85 years. It shows that the population is more prone to be the elderly group. This may affect the findings as the health status of the elderly may not be suitable or representative enough to the 18-85 years population. The gender subgroup analysis is also carried out but basically the age is not taking into consideration.
The basic policy was to include adults without any age bias, which is why the inclusion criterium was >18 not to provide any doubts in this context. Of course, we were ready to include any adult patient with CAD, however, in clinical practice, patients with CAD and the ones undergoing coronary angiography are usually older than 50 years. In general, it is rare for CAD to occur before 50 years. This explains the deceptive age shift of the included participants compared to the inclusion criteria. Indeed, the youngest participant in this study was 50 years old, which is in line which the general characteristics of the CAD course of the disease. If we would analyze the general CAD population (50-85 y.o.) then the mean age in this study (67.5y.o.) is exactly the mean age of the CAD population.
- Table 1,3 & 4, the values for CI is slightly confusing, the authors may put a bigger gap between or put a dash to indicate the interval.
It was corrected to make it easier to read. Dash would be confusing as some values are minus ones, that is why semicolon was used to separate these values.
Reviewer 2 Report
Dear authors,
Thanks for the study, the idea of assessing the nutritional status of coronary artery disease patients is interesting and relevant, but I have some suggestions:
1. Please make corrections in the abstract, see the corrections and comments made in the publication.
2. It would be advisable to give recommended/optimal values for each indicator (Lines 63-65) so that the results of the study can be better understood.
3. Move Lines 165-174 to the section Materials and Methods (after Line 103).
4. See the corrections and comments made in the publication.

Author Response
Dear Reviewer,
Thank you very much for your time, effort, and valuable comments on our manuscript. We corrected the manuscript according to your suggestions and we believe that it deeply improved its quality. Below you can find our responses to your comments. Thank you for this opportunity.
Dear authors,
Thanks for the study, the idea of assessing the nutritional status of coronary artery disease patients is interesting and relevant, but I have some suggestions:
Please make corrections in the abstract, see the corrections and comments made in the publication.
The corrections were made according to the comments in the pdf file.
It would be advisable to give recommended/optimal values for each indicator (Lines 63-65) so that the results of the study can be better understood.
The normal range values were added according to the suggestions in the pdf file.
Move Lines 165-174 to the section Materials and Methods (after Line 103).
The lines were moved as suggested.
See the corrections and comments made in the publication.
The corrections were made according to the comments in the pdf file (unnecessary table footers were deleted).
Reviewer 3 Report
Dear authors,
I have noticed some points that must be corrected:
1. Line 30 "ESPEN"- it is not provide explanation of the abbreviation.
2. Line 43 "According to ESPEN guidelines, malnutrition can be also diagnosed with this 43 parameter [1]."- explain better. The sentence is not clear.
3. The conclusion must be revised. Which nutritional status is the better one?
Author Response
Dear Reviewer,
Thank you very much for your time, effort, and valuable comments on our manuscript. We corrected the manuscript according to your suggestions and we believe that it deeply improved its quality. Below you can find our responses to your comments. Thank you for this opportunity.
Dear authors,
I have noticed some points that must be corrected:
Line 30 "ESPEN"- it is not provide explanation of the abbreviation.
The full name was added
Line 43 "According to ESPEN guidelines, malnutrition can be also diagnosed with this 43 parameter [1]."- explain better. The sentence is not clear.
The sentence was corrected to make it clear.
The conclusion must be revised. Which nutritional status is the better one?
The conclusions were revised. NRS 2002 and BIA are the tools that analyze different aspects, thus we cannot define “a better one”, however, we can suggest that due to that they are possibly complementary. Malnutrition is worse for patients in terms of CAD performance, which was emphasized in the conclusions section.
Round 2
Reviewer 3 Report
Dear authors,
I have read the revised version of your manuscript.
However, there are some issues that have to be revised:
1. Line 98- you have used different fonts. Please check carefully the "instructions to authors" of MDPI.
2. Line 175 and 176- "Finally, the patients were divided into subgroups by sex (men and women) in which 175 the mentioned analyses were also performed."
Please describe these groups.
3. The study design is not clearly enough described- my suggestion is to include a figure/ graphique for better explanation.
4. Did you had any data about the nutritional status of the participants in the study? For example were the participants vegans, vegetarians etc.? If you have such data, you can include it in the manuscript.
4. Informed Consent Statement- you can included it in the supplementary data.
Lines 376-377 "Informed Consent Statement: Informed consent was obtained from all patients involved in the 376 study.."
The sentence ends with 2 points. Correct it.
5. The conclusion is too short.
Author Response
Dear Reviewer,
Thank you very much for your time, effort, and valuable comments on our manuscript. We corrected the manuscript according to your suggestions and we believe that it deeply improved its quality. Below you can find our responses to your comments. Thank you for this opportunity.
Dear authors,
I have read the revised version of your manuscript. However, there are some issues that have to be revised:
1. Line 98- you have used different fonts. Please check carefully the "instructions to authors" of MDPI.
The used font throughout the whole manuscript was Palatino, as indicated in the instructions for authors and as included in the MDPI IJERPH MS Word Template. However, the whole manuscript was checked again for proper formatting as suggested by the Reviewer.
2. Line 175 and 176- "Finally, the patients were divided into subgroups by sex (men and women) in which 175 the mentioned analyses were also performed." Please describe these groups.
The mentioned sentence was originally in line 185. A detailed description of the subgroups was additionally provided in lines 186-194.
3. The study design is not clearly enough described- my suggestion is to include a figure/ graphique for better explanation.
The main study design was presented in the figure as Graphical abstract.
4. Did you had any data about the nutritional status of the participants in the study? For example were the participants vegans, vegetarians etc.? If you have such data, you can include it in the manuscript.
Additional data about such types of dietary patterns (e.g. vegan etc.) was not collected from patients. They were asked about general intake changes, however, this aspect is required for NRS 2002 assessment. Thank you very much for this kind suggestion, we will examine that relationship in our future projects.
5. Informed Consent Statement- you can included it in the supplementary data.
We can include the form (translated from Polish to English) if you find it necessary, however, it is unusual to attach a consent form to the manuscript, which is why it was not attached to the manuscript. The consent (originally in Polish) was:
“I, the undersigned, declare that I voluntarily and knowingly consent to participate in the study. I have read the information for the study participant, I had the opportunity to ask questions to the person conducting the study and I received answers to them. I am aware that I can revoke my consent and resign at any time from participating in the research without giving a reason.”
6. Lines 376-377 "Informed Consent Statement: Informed consent was obtained from all patients involved in the 376 study.." The sentence ends with 2 points. Correct it.
It was corrected as suggested by the Reviewer.
7. The conclusion is too short.
The conclusion section was extended as suggested by the Reviewer.